# ChimpanSEE, ChimpanDO: Grooming and play contagion in chimpanzees

**Georgia Sandars** [ID]*, **Jake S. Brooker** [ID], **Zanna Clay** [ID]*

Department of Psychology, Durham University, Durham, United Kingdom

* georgia.sandars@durham.ac.uk (GS); zanna.e.clay@durham.ac.uk (ZC)

**Data Availability Statement:** All relevant data files are available from the figshare database: https://doi.org/10.6084/m9.figshare.25314394.v1 (DOI: 10.6084/m9.figshare.25314394).

## Abstract

Behavioural contagion—the onset of a species-typical behaviour soon after witnessing it in a conspecific—forms the foundation of behavioural synchrony and cohesive group living in social animals. Although past research has mostly focused on negative emotions or neutral contexts, the sharing of positive emotions in particular may be key for social affiliation. We investigated the contagion of two socially affiliative interactive behaviours, grooming and play, in chimpanzees. We collected naturalistic observations of $N = 41$ sanctuary-living chimpanzees at Chimfunshi Wildlife Orphanage, conducting focal follows of individuals following observations of a grooming or play bout, compared with matched controls. We then tested whether the presence and latency of behavioural contagion was influenced by age, sex, rank, and social closeness. Our results offer evidence for the presence of grooming and play contagion in sanctuary-living chimpanzees. Grooming contagion appeared to be influenced by social closeness, whilst play contagion was more pronounced in younger individuals. These findings emphasise that contagion is not restricted to negatively valenced or self-directed behaviours, and that the predictors of contagious behaviour are highly specific to the behaviour and species in question. Examining the factors that influence this foundational social process contributes to theories of affective state matching and is key for understanding social bonding and group dynamics.

## Introduction

Our social environments are shaped by our ability to understand and respond to the states and behaviours of those around us. A foundational form of sensitivity to others is behavioural contagion—the onset of a species-typical behaviour soon after witnessing it in a conspecific [1]. This phenomenon is found across many animal species, including our primate relatives, and is considered to play an important role in group cohesion and social living [2].

Behavioural contagion is intimately linked to emotional contagion and other socio-cognitive processes such as empathy and social learning [3–5]. Studies of behavioural contagion have classically focused on the spread of actions associated with displeasure or negative stimuli, including self-directed behaviour e.g., scratching; [6] and non-interactive behaviour e.g., vigilance; [7]. These have been suggested to provide immediate survival benefits [7]. 'Positive' behavioural contagion includes the contagion of affiliative social behaviours associated with

**Funding:** This work was funded by the Templeton World Charity Foundation (grant number 0309 to Z.C., http://www.templetonworldcharity.org). The funders had no role in study design, data collection and analysis, decision to publish, or preparation of the manuscript.

**Competing interests:** The authors have declared that no competing interests exist.

pleasure which may assist animals to develop and strengthen cooperative social bonds [8]. Contagion of affiliative behaviours has comparatively received less empirical attention.

Within primates, affiliative bonds can be developed and maintained through grooming and play interactions [9,10]. Grooming is combing through the fur of oneself (autogrooming) or a partner (allogrooming) to remove dirt, foreign objects or parasites [11], which, in addition to physical benefits such as fur cleaning and parasite removal [12], is purported to mediate social tension [13] and has been associated with pleasurable emotions [14]. Furthermore, grooming partners are more likely to mate, share food, and protect one another's infants [15]. Whilst infant primates are often groomed by adults, typically their mothers, it is adults that typically benefit from the social bonding implications of allogrooming [9]. In contrast, young primates typically form their first non-maternal relationships through social play, although social play is also found in adults [10,16]. Play includes a wide variety of activity; both behaviours that are reminiscent of serious functional contexts (e.g., fighting and mating behaviour), and of actions that have no immediate function or benefit (e.g. somersaults) [17]. Play can be identified by typical action patterns and in some primates, social play is associated with a 'play face' which signals playful intentions to the receiver [18]. Whilst solitary play is common, social play (hereafter: "play") involves interaction with another individual.

Although play is generally considered a positive interaction, there is no clear link between play and positive affect [19]. In adult chimpanzees, there are key structural differences between play fighting and real fighting, but play may still be used as an alternative to aggression, to establish or maintain dominance relationships [20] and so should not be considered a wholly positive behaviour. Play is also important for enabling animals to learn about others' specific behavioural tendencies, and to practise particular skills and motor patterns useful for future interactions [10,21,22]. More broadly, play is used to establish and maintain social relationships, integrating juveniles into their wider communities [23].

In primates, contagion of social behaviours was first observed in female Barbary macaques, who were quicker and more likely to initiate grooming after observing others groom [8]. This contagion was viewed to be 'positive' due to a simultaneous increase in other affiliative behaviours, and a decrease in behavioural indicators of anxiety, after individuals observed grooming. This is suggestive of a more general transmission of emotional state rather than just behaviour. Grooming contagion has also been identified in female rhesus macaques, appearing to be more pronounced in higher ranked observers, but not influenced by social closeness [24]. Whilst play contagion has not been directly observed in primates, young ravens are reported as more likely to play after observing others play [25]. Ravens did not necessarily engage in the same type of play they observed, which indicates a more general behavioural or emotional contagion, as opposed to motor mimicry. A study of play contagion has also been carried out with calves, finding a negative contagion effect, whereby play was supressed when exposed to others who played less [26]. Many other behaviours have also been shown to be contagious- including vigilance in Japanese macaques [7], scent-marking in marmosets [27], and self-scratching and yawning in many different species for review, [3].

As our closest living relatives, chimpanzees share many socio-emotional processes and behaviours with humans; they invest in long-term social relationships, live in similar social structures and rely on similar social bonding strategies. This makes chimpanzees a suitable model for studying open questions on the origins of human social processes. Further research on social contagion can enable cross-species comparisons with humans and other primates, which are key to detecting socio-cognitive and behavioural differences, and marking shifts in the evolution of hominin sociality.

Chimpanzee behaviour and emotional states have been shown to be influenced by those around them. Observational and experimental studies of yawn contagion have shown that

chimpanzees are sensitive to the states of others, for review: [3] with some indications that increased familiarity may influence the strength of the contagion effect [28; but see 29,30]. Chimpanzees also demonstrate mimicry, the involuntary, automatic and fast copying of a single component motor action [31]. Chimpanzees across all age classes, sexes, and rank classes quickly replicate the play faces of others, which is purported to modulate play sessions and communicate playful intentions [18]. Two further studies have shown that affiliative and agonistic social behaviours have been shown to spread in chimpanzees via vocal contagion. In captive chimpanzees, hearing grooming vocalisations from neighbouring groups led to an increase in grooming [32], whilst hearing agonistic vocalisations led to increased aggressive displays and vocalisations [33]. As this does not just involve matching the behaviour observed, but a cross-modal response, this suggests that it involves a higher order associative process, or perhaps that the contagion of emotions is driving the effect.

Overall, there is some evidence that positive behaviours are contagious in other species, although this has not been addressed in chimpanzees, who have been found to exhibit a variety of other contagious processes. Therefore, we sought to establish whether chimpanzees show contagion for two purportedly positively-valenced affiliative behaviours—grooming and play—and to establish what influences the presence and latency of the contagion effect.

We first tested the hypothesis that chimpanzees would exhibit behavioural contagion of grooming and play.

As chimpanzees display related contagious processes, and are highly socially aware and sensitive to the emotional expressions and behaviours of their peers [3,28,32], we expected to find evidence of contagion.

We predicted that chimpanzees would initiate grooming more frequently after having just observed others groom, and initiate play more frequently after having just observed others play.

We then investigated whether individual and social factors would influence the likelihood of play and grooming contagion, and the latency at which contagion occurs, focussing on social closeness, sex, age, and dominance rank. When observer and stimulus individuals are close social partners, there is some evidence for increased facial mimicry [34] and increased yawn contagion [35,36; but see 29,30]. This is thought to be either due to an attention bias towards socially close individuals (the '*Attention Bias Hypothesis*'), or due to increased emotional transfer between close individuals, as found in empathy [the '*Emotional Bias Hypothesis*'; 3,37]. We therefore predicted that grooming and play contagion would be more prevalent, and responses would occur quicker, between close social partners.

Individual characteristics may also determine the presence and latency of the contagion effect, and age is an often studied variable. Research on facial mimicry and yawn contagion indicates that contagion mechanisms are present from infancy in chimpanzees, although some studies have found stronger effects in older individuals for review: [3]. Contagion of affiliative behaviours may show distinct patterns, as initiating a behaviour is a multicomponent process involving social cognition, and its expression is under voluntary control. As impulse control and executive function increases over adolescence in primates [38,39], we predicted that the contagion effect for grooming and play would be more pronounced in younger chimpanzees.

We also considered the effect of rank on contagion. Chimpanzees have a broadly linear hierarchy in which lower ranking individuals are more constrained in their actions [40]. In macaques, it was found that lower ranking individuals displayed less behavioural contagion, which may be due to a greater inhibition of behavioural responses because of their low social mobility [24]. We would expect that similarly, high ranking chimpanzees would be more able to act freely, and therefore we predicted they would show increased and faster grooming and play contagion. Finally, we considered the effect of sex on contagion. As female and male

**Table 1. Social composition of the study population from Chimfunshi Wildlife Orphanage as of the start of our observations (29/05/2021).**

|  | Females | Males | Total |
|---|---|---|---|
| Infants (0–2 yrs) | 6 | 5 | 11 |
| Juveniles (3–7 yrs) | 6 | 4 | 10 |
| Subadults (8–11 yrs) | 1 | 4 | 5 |
| Adults (12+ years) | 23 | 9 | 32 |
| Total | 36 | 22 | 58 |

chimpanzees may follow different social bonding strategies, and different communities demonstrate different patterns [41–43], we formed a non-directional hypothesis that sex differences would be evident in the contagion of affiliative social behaviours.

## Methods

### Study site and subjects

We observed a community of $N = 58$ chimpanzees at Chimfunshi Wildlife Orphanage, a sanctuary site in the Copperbelt Province of Zambia. Our subjects reside in a 160-acre enclosure connected to an indoor handling facility. Compared to other communities at the sanctuary, the group we studied ('Group 2') has been reported to be a relatively stable and tolerant social group [44,45]. A demographic breakdown of sex and age classes for the study group is shown in Table 1.

We logged approximately 237 observation hours between 29th May and 31st July 2021. On each day, we observed the chimpanzees from 08:00 to 11:00 and 14:00 to 17:00, outside of the regulated feeding period (when chimpanzees were provisioned with supplementary food and thus artificially encouraged to congregate). We observed all chimpanzees in the group who were 3-years-old or older; we excluded individuals younger than 3-years-old due to mother dependency, lack of independent social connections [46], and markedly low tendencies to initiate grooming [47]. Some individuals ($N = 6$) spent most of their time deeper inside the enclosure where they were out of sight, meaning we were not able to collect enough data for them to include in the final analyses. This resulted in an overall sample of $N = 41$ chimpanzees.

### Data collection

**Post-observation / matched control (PO-MC) focals.** We collected behavioural data using a procedure adapted from the post-conflict/matched control (PC-MC) method, developed by de Waal and Yoshihara [48] to study post-conflict behaviour. The PC-MC method has recently been applied to study behavioural changes post-observation of grooming [8,24]. The PC-MC method involves recording the behaviour of an individual for a set amount of time immediately after they observe a conflict, and comparing this to behaviour recorded during a control period of time, in which they have not just observed a conflict, but where other conditions are matched. In this study, we collected data for the 5-minute period after an individual either observed grooming or observed play.

We collected all post-observation (PO) focals opportunistically, from the start of when an individual observed either grooming or play. We determined whether observation happened by considering the head orientation of the observer, and their distance to the behaviour. When the event was within 5-metres, and happened in the 180˚ in front of the observer's face and in direct visual contact, we considered the behaviour as observed. PO focals therefore started either when a grooming or play interaction started within the subject's observational area, or when a subject

moved so that they observed the behaviour. Individuals who participated in the first play/grooming bout, even if they did not initiate it, were not included as observers, as they were already involved in the behaviour. For each PO (play or grooming), we then followed the observer for 5-minutes, recording all behaviour using a handheld digital video-camera (Panasonic HC V777) and detachable directional microphone (Sennheiser MKE 400). During these focal follows, we narrated the IDs of all other chimpanzees present, and which were visible to the focal, and noted all grooming and play interactions that the focal subject observed or engaged in.

We also recorded control focals opportunistically, selecting a focal individual at random if there were multiple available. We ensured that the focal chimpanzee had not just observed play or grooming by following them for 5-minutes before starting the focal, and we ensured that there were at least two individuals present within a 5-metre radius, so they had the opportunity to engage in social play or grooming if they wanted. We then followed the focal chimpanzee for 5-minutes as with the post-observation focals, narrating relevant behaviour. This more flexible adaptation of the original De Waal & Yoshihara [48] method enabled us to control for the number of surrounding individuals, as well as being more practical to implement in a sanctuary environment where the chimpanzees were not regularly visible.

Data was originally collected by randomly selecting one of multiple possible individuals to follow if there were multiple options. In the final two weeks of data collection, we prioritised individuals for whom there was the lowest amount of focals collected.

**Social affiliation data.** To assess dyadic social relationship strength, we conducted instantaneous scan samplings at 5-minute intervals during non-feeding periods, approximately between 07:00–11:00 and 14:00–17:00 [49]. For each scan, we recorded the identities of all individuals present and all social interactions. Specifically, we recorded instances of grooming, play, contact sitting, and proximity sitting (< 1-metre) once per dyad per scan. Scans were visually recorded using Panasonic VC-777 video cameras.

To compute the social closeness of each dyad, we used the social scan data to compute a dyadic sociality index (DSI; [24,50]. We calculated the proportion of time that each dyad spent engaging in each interactive behaviour (playing, grooming, contact sitting, proximity sitting), by dividing the number of scan-points they engaged in the behaviour by the total number of scan-points where at least one chimpanzee in the dyad was in view. The metrics of each behaviour correlated, and so we integrated them into an overall DSI. We divided each metric by the average of that metric across all dyads, and averaged the 4 scores for each individual [24,50]. The DSI scores were entered into the GLMMs.

We also used this social data to compute scores for each individual's overall grooming tendency and play tendency, by dividing the number of scan-points they respectively groomed and played in, by the total number of scans the individual was present for. These grooming/play tendency scores were entered into the GLMMs.

**Individual characteristics data.** The age and sex of individuals was determined using veterinary records; birth dates are recorded for mother-reared individuals and estimated upon arrival for those born in the wild. We assigned a linear dominance rank hierarchy according to deliberated agreement between four long-term experienced keepers responsible for daily care and food provision. This ranking was based on experience watching dyadic agonism and patterns of submission during feeding times. This method has been used in previous research looking at the influence of rank on behaviour, carried out at Chimfunshi Wildlife Orphanage [51].

## Data coding

We conducted all-occurrence coding of affiliative interactions during PO and MC focals [46], using ELAN [version 5.9, 52]. ELAN facilitates precise recording of the onset and duration of

behaviours in observational research. We applied a systematic behavioural ethogram in our coding protocols (see supporting information) based on previous grooming [32], play [53,54], and gestural studies [55,56].

We coded the occurrence of contagion as a binomial "yes/no" (1/0) variable. If another individual initiated a grooming or play bout with the focal before the focal initiated a behaviour, this post-observation follow was excluded from the GLMM analyses, as it would not be possible to determine whether the focal's subsequent behaviour was driven by their experience observing or engaging in the behaviour. In a minority of focals, grooming or play was initiated by the focal individual multiple times. In these cases, we only consider the first grooming or play interaction they engaged in, as similarly, it would not be possible to determine whether this behaviour was influenced by the focal's initial observation or ensuing engagement in the behaviour. Latency (i.e., duration between moment of observation and initiating a matching behaviour) was calculated and expressed as a proportion of the 5-minute focal follow for the purpose of analysis in the GLMMs.

To check inter-coder reliability, we allocated a subset (15%) of video data, balanced across grooming PO, play PO, and MC focals, for secondary coding by two independent observers. Intercoder reliability mean dyadic agreement was established using Cohen's Kappa values, to determine the presence and initiation of all behaviours that were entered into the models. Kappa values all exceeded 0.8, indicating very good agreement [57].

## Data analysis

**Question 1: Is there a behavioural contagion effect?.**   In order to assess presence of behavioural contagion, each post-observation follow ("PO") was paired with a matched control ("MC"). The videos were matched in a way that maximised the number of pairs matched by time of day (within 1 hour of each other), and then also by number of individuals present if possible (19% of videos), or if not by the approximate (+/- 2) number of individuals.

Following de Waal and Yoshihara [48], a PO-MC pair was counted as 'attracted' if the focal individual initiated the relevant behaviour in the PO but not MC, 'dispersed' if the focal individual initiated the behaviour in the MC but not the PO, and 'neutral' if the behaviour was initiated in both or in neither.

We excluded POs and MCs where another individual initiated grooming which the focal engaged in, before the focal had initiated grooming.

To detect evidence of a contagion effect (indicated by attracted pairs), we compared rates of attracted, neutral, and dispersed pairs of focals, for the grooming videos and play videos separately. We used a Friedman test, and then conducted post-hoc pairwise comparisons using Wilcoxon tests with Bonferroni corrections, running all analyses in RStudio (Version 4.2.2).

**Question 2: Which factors moderate the contagion effect?.**   We fitted four General Linear Mixed Models (GLMMs) to test which individual and social characteristics influenced grooming and play contagion. We tested which variables predicted whether grooming was initiated post observation (Model 1.1) and the latency until grooming was initiated (Model 1.2), and which variables predicted whether play was initiated post observation (Model 2.1) and the latency until play was initiated (Model 2.2).

As fixed effects, we included predictor variables of age, sex, and rank of the focal, and the social closeness between the focal and the individuals they observed. We included four control effects: time of day, overall tendency to groom/play (groom for Models 1.1–1.2, play for Models 2.1–2.2), number of other individuals present, and number of grooming/play bouts observed. We also included a crossed random effects structure consisting of random intercepts for the focal ID and event ID (as some observations were recorded during the same bout of

grooming or play). In Models 1.1 and 2.1, event ID de-stabilised the model and caused convergence issues, due to the majority of datapoints not having a repeated event ID (110 of 159 events in Model 1.1, and 82 of 136 events in Model 2.1). Event ID was not a significant predictor, and its exclusion resulted in a negligible reduction of model log-likelihood, and so we excluded it from these models.

In Models 1.1 and 2.1, we included all theoretically identifiable random slopes and correlation terms where relevant, in order to prevent a type 1 error and avoid overconfidence in terms of precision of fixed effects estimates [58]. In Model 1.1, we included the random slopes of social closeness within focal ID and number of others present within focal ID; in Model 2.1 there were no theoretically identifiable slopes. In Models 1.2 and 2.2, due to a limited dataframe, we did not include any random slopes to prevent overcomplexity.

We fitted all models in RStudio (Version 4.2.2). For Model 1.1 and Model 2.1, looking at whether or not the focal initiated the behaviour, we used a GLMM model with Binomial error structure and logit link function [59] and tested this using the function lmer from the package lme4. For Model 1.2 and Model 2.2, looking at the latency until the behaviour was initiated, we used a GLMM with a beta error structure and a logit link function [60,61] and tested this using the function glmmTMB from the package glmmTMB.

Before fitting each model, we inspected the distribution of all the covariates, to check that they were roughly symmetrical and free of outliers. We log transformed covariates that were skewed (social closeness, and number of grooming/play bouts observed), and then z-transformed all covariates, to ease model convergence and interpretability. We assessed model stability with a function comparing estimates obtained from full models based on all data with those obtained from models with the levels of the random effects excluded one at a time. Confidence intervals were derived using the boot function of the package lme4 using 1,000 parametric bootstraps and bootstrapping over the random effects too [62].

In order to assess the overall strength of the model without cryptic multiple testing [63], we then conducted full-null model comparisons, comparing each full model with a null model which lacked the fixed effects but included all control effects and random effects, using likelihood ratio tests [64]. For each model, we then tested the effect of individual fixed effects (age, sex, rank and social closeness) by comparing the full model with reduced models which dropped the fixed effect terms one at a time using drop1 tests [58].

## Ethics

This study comprises of entirely naturalistic, non-invasive observational methods, strictly adhering to all legal requirements of Zambia, as well as the International Primatological Society's Principles for the Ethical Treatment of Nonhuman Primates. This study was approved by the Animal Welfare Ethical Review Board (AWERB) of Durham University and the Chimfunshi Research Advisory Board (CRAB).

## Results

### Question 1: Is there a behavioural contagion effect?

**Grooming.** To run the analyses, we included individuals for whom there were at least two matched pairs, in order to generate a proportion of attracted/dispersed pairs between 0 and 1. This resulted in a total of $N = 120$ PO-MC pairs (for $N = 32$ individuals), of which $N = 29$ were attracted, $N = 1$ was dispersed, and $N = 90$ were neutral (proportions per individual are shown in Fig 1). A Friedman test revealed that, across individuals, there was a significant difference in rates of attracted, neutral, and dispersed pairs ($N = 32$, $\chi^2 = 45.6$, $W = 0.72$, $P < .001$). We conducted post-hoc pairwise comparisons using Wilcoxon tests with Bonferroni corrections,

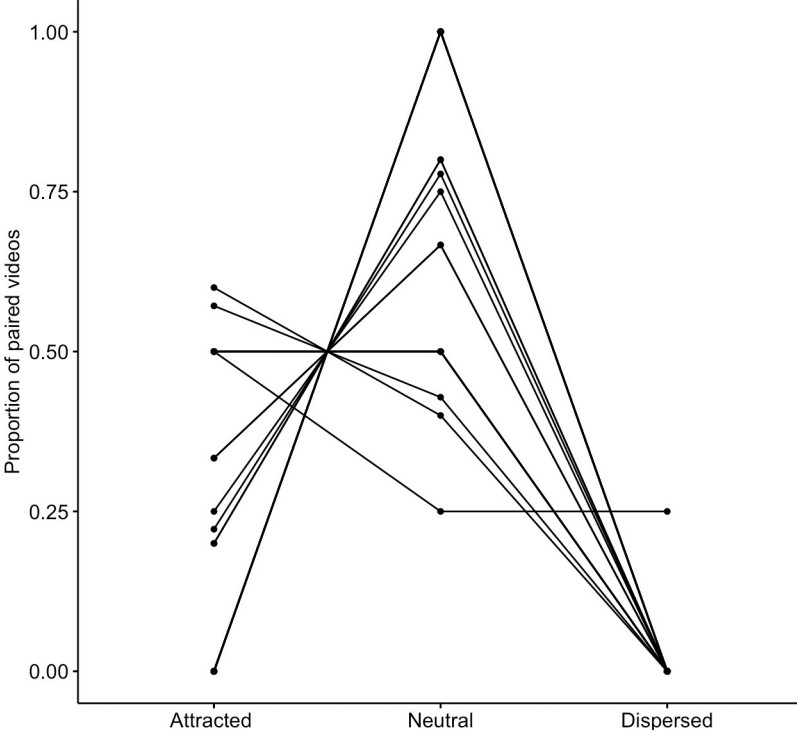

**Fig 1. Proportions of attracted, neutral, and dispersed grooming PO-MC pairs, for each individual.** Attracted pairs correspond to grooming happening only in the PO, dispersed pairs correspond to grooming happening only in the MC, and neutral pairs correspond to grooming happening in neither or both. Data comprised of $N$ = 120 matched focals of $N$ = 32 individuals.

which showed that, across individuals, the average proportion of attracted PO-MC pairs was significantly higher than the average proportion of dispersed pairs ($N$ = 32, $Z$ = 3.41, $r$ = 0.68, $P$ = .002).

This indicates it was more common for individuals to initiate grooming in the PO and not in the MC than in the MC and not in the PO, providing evidence for a grooming contagion effect. Additionally, there were significantly higher rates of neutral pairs than attracted pairs ($N$ = 32, $Z$ = 4.44, $r$ = 0.88, $P$ < .001), and of neutral rates than dispersed pairs ($N$ = 32, $Z$ = 4.95, $r$ = 0.99, $P$ < .001).

**Play.** To run the analyses, we again only included individuals for whom there were at least two matched pairs. This resulted in a total of $N$ = 96 PO-MC pairs (*for N* = 25 individuals), of which $N$ = 33 were attracted, $N$ = 1 was dispersed, and $N$ = 62 were neutral (proportions per individual are shown in Fig 2). A Friedman test revealed that, across individuals, there was a significant difference in rates of attracted, neutral, and dispersed pairs ($N$ = 25, $\chi^2$ = 33.0, $W$ = 0.66, $P$ < .001). We conducted post-hoc pairwise comparisons using Wilcoxon tests with Bonferroni corrections, which showed that, across individuals, the average proportion of attracted PO-MC pairs was significantly higher than the average proportion of dispersed pairs ($N$ = 25, $Z$ = 3.81, $r$ = 0.76, $P$ < .001). This indicates it was more common for individuals to initiate play in the PO and not in the MC than in the MC and not in the PO, providing evidence for a play contagion effect. Additionally, there were significantly higher rates of neutral pairs than dispersed pairs ($N$ = 25, $Z$ = 4.26, $r$ = 0.85, $P$ < .001), but no significant difference between neutral pairs and attracted pairs ($N$ = 32, $Z$ = 2.19, $r$ = 0.44, $P$ = .085).

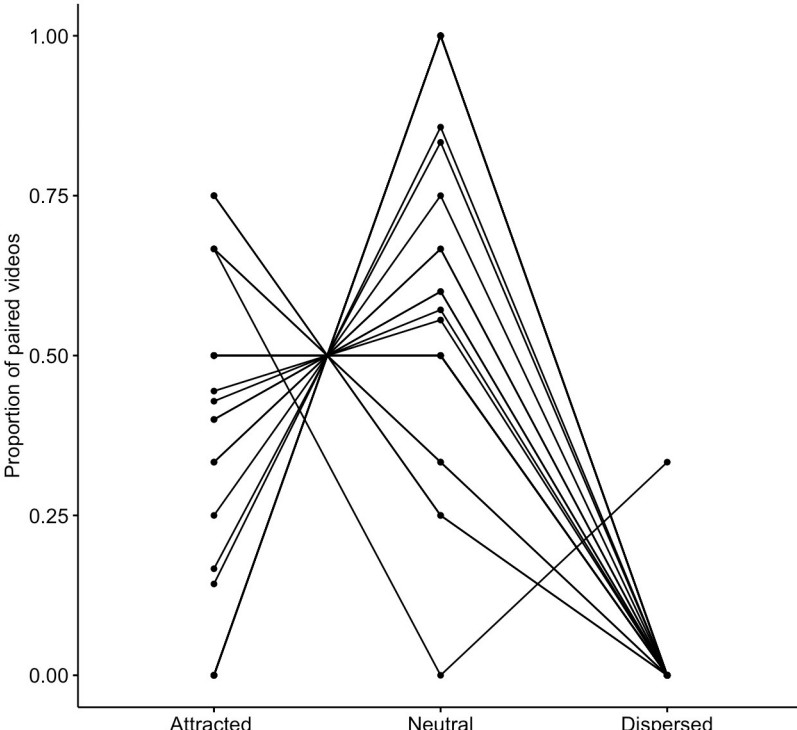

**Fig 2. Proportions of attracted, neutral and dispersed play PO-MC pairs, for each individual.** Attracted pairs correspond to play happening only in the PO, dispersed pairs correspond to play happening only in the MC, and neutral pairs correspond to play happening in neither or both. Data comprised of $N = 96$ matched focals of $N = 25$ individuals.

## Question 2: Which factors moderate the contagion effect?

Full model results, including contributions of all fixed and random effects, can be found in the supporting information.

**Grooming contagion occurrence (Model 1.1).** The final data set analysed comprised $N = 159$ observations, of $N = 40$ individuals (1–12 observations per individual). We modelled the effect of four predictor variables (age, sex, rank, social closeness) and four control variables (time of day, grooming tendency, number of other individuals present, number of grooming bouts observed) on the dependent variable of presence of grooming initiation. Overall, the full model provided a significantly better fit than the null model ($\chi^2 = 14.013$, df = 4, $P = .007$). There was a significant positive effect of social closeness ($\chi^2 = 5.904$, df = 1, $P = .015$) whereby, as shown in Fig 3, the stronger the dyadic relationship between the observer and stimulus individuals, the more likely the observer was to initiate a grooming bout. Age, sex, and rank were not significant predictors. All model estimates appeared relatively stable.

**Grooming contagion latency (Model 1.2).** The final data set analysed comprised of $N = 54$ observations, of $N = 21$ individuals (1–6 observations per individual), across $N = 41$ events (1–3 observations per event). We modelled the effect of four predictor variables (age, sex, rank, social closeness) and four control variables (time of day, grooming tendency, number of other individuals present, number of grooming bouts observed) on the dependent variable of latency until grooming initiation. The full model was not a significantly better fit than the null model ($\chi^2 = 4.986$, df = 4, $P = .288$), and so results should be treated with caution. All

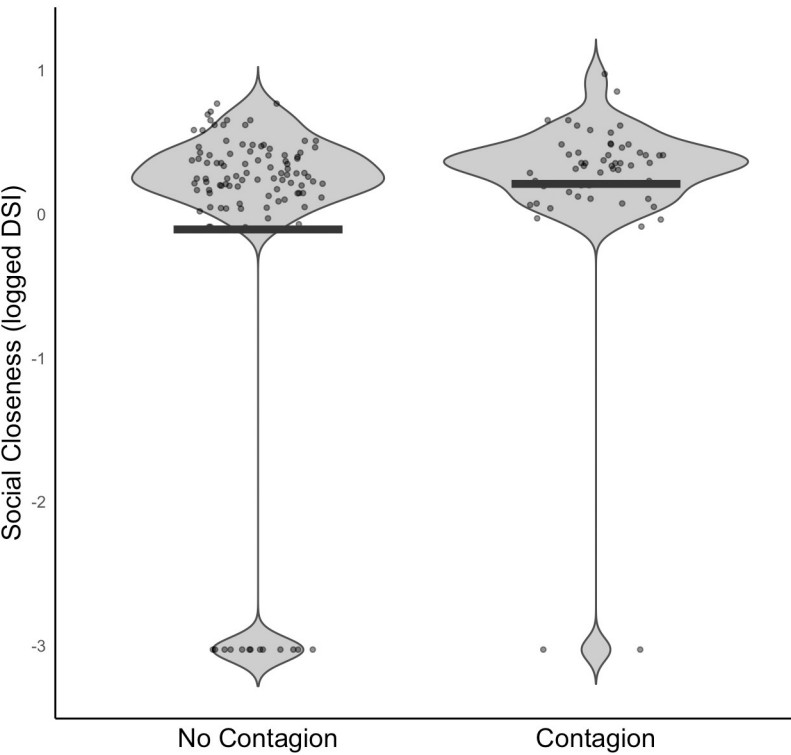

**Fig 3. Social closeness scores in focal follows where contagion did and did not take place.** The social closeness measure is the z-transformed dyadic sociality index between observer and stimulus individuals. The mean social closeness value for instances of no contagion vs contagion is shown with a black bar. Data comprised of $N = 159$ observations of $N = 40$ chimpanzees.

model estimates appeared relatively stable. Results showed that post-observation grooming was not immediate, although grooming peaked within the first minute (Fig 4). There was a median latency of 77.1 seconds, lower quartile of 24.5 seconds, and upper quartile of 119.5 seconds.

**Play contagion occurrence (Model 2.1).** The final data set analysed comprised of $N = 136$ observations, of $N = 41$ individuals (1–12 observations per individual). We modelled the effect of four predictor variables (age, sex, rank, social closeness) and four control variables (time of day, play tendency, number of other individuals present, number of play bouts observed) on the dependent variable of presence of play initiation. The full model was a significantly better fit than the null model ($\chi^2 = 14.873$, df = 4, $P = .005$). We found a significant effect for age ($\chi^2 = 11.461$, df = 1, $P = .001$), indicating that younger focals were more likely to initiate play after observation, as shown in Fig 5. We did not find a significant contribution of sex, rank, or social closeness. All model estimates appeared relatively stable.

**Play contagion latency (Model 2.2).** The final data set analysed comprised of $N = 48$ observations, of $N = 26$ individuals (1–5 observations per individual), across $N = 37$ events (1–3 observations per event). We modelled the effect of four predictor variables (age, sex, rank, social closeness) and four control variables (time of day, play tendency, number of other individuals present, number of grooming bouts observed) on the dependent variable of latency until play initiation. The full model was not a significantly better fit than the null model ($\chi^2 = 5.212$, df = 4, $P = .266$), and so results should be treated with caution. All model estimates appeared relatively stable. Our analysis revealed that post-observation play was most likely to

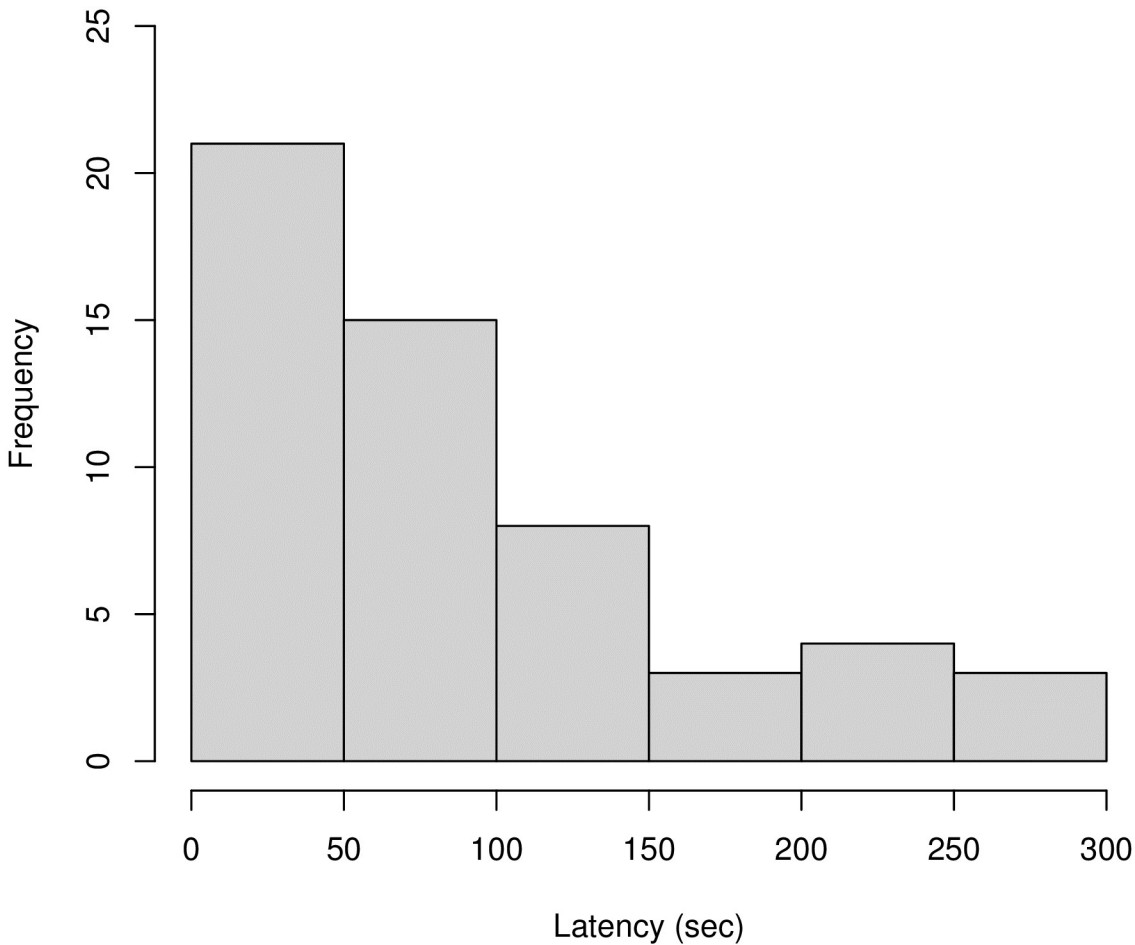

**Fig 4. Histogram of latencies from observing grooming to first initiating grooming.** Data comprised of *N* = 54 observations across *N* = 21 individuals.

occur within the first minute, as shown in Fig 6. There was a median latency of 18.0 seconds, lower quartile of 3.5 seconds, and upper quartile of 45.4 seconds. Having observed that play data latencies appeared much shorter than the grooming data latencies, we carried out an exploratory Welch's T-test, to compare latencies until the initiation of behaviour post-observation. We found that latencies for the play data were significantly shorter than the latencies for the grooming data (t = 2.758, df = 99.9, p < .001).

## Discussion

Here we report evidence of grooming and play contagion in sanctuary-living chimpanzees. When exposed to the social interactions of others, sanctuary-living chimpanzees also show significant tendencies to catch the behaviours that they observe. We found evidence for a contagion effect for both grooming and play, and this effect was significant for a sample including all ranks and sexes, and a wide age range. Our findings extend our understanding of behavioural contagion in our closest living relatives, whereby not only do chimpanzees catch yawns from one another [3], but they appear to also catch affiliative social behaviours.

Consistent with the hypothesis that behavioural contagion represents a basal layer of empathy [4,65], grooming contagion was more likely between close social partners. A heightened

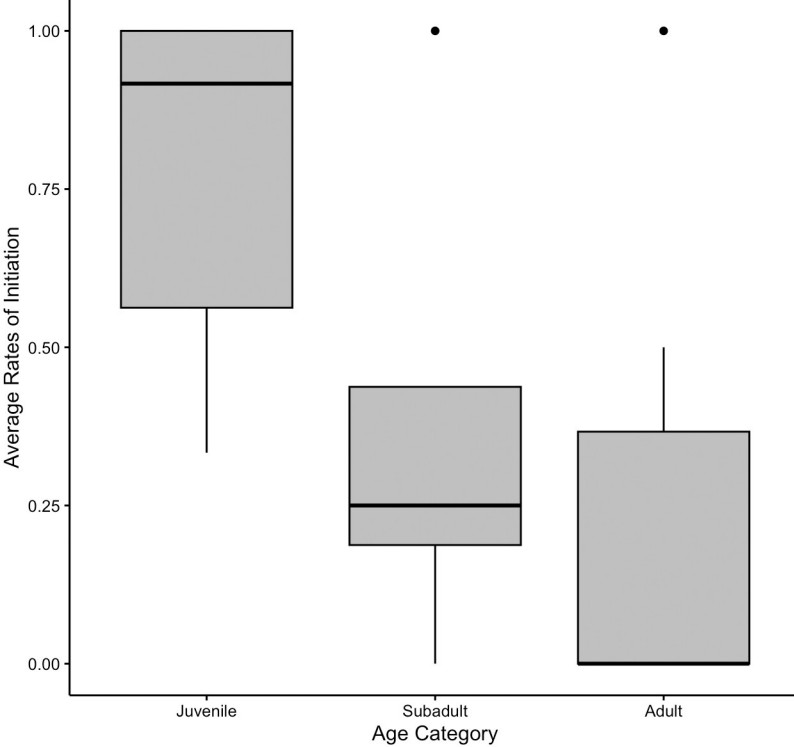

**Fig 5. Mean rate of post-observation play across age categories.** Data is grouped into juveniles (3–7 years; $N = 34$ observations of $N = 10$ chimpanzees), subadults (8–11 years; $N = 20$ observations of $N = 4$ chimpanzees) and adults (12 + years; $N = 82$ observations of $N = 27$ chimpanzees). This categorisation is for visualisation purposes, but data were analysed with age as a continuous variable. Upper and lower quartiles are indicated by the box boundaries, and dots indicate outliers.

effect between socially close individuals has previously been reported in primate studies of yawn contagion and facial mimicry [34,35], as well as in many empathy studies [e.g., 66], although a recent scratch contagion study found an opposite effect [67]. This social closeness bias could be explained either by an attention bias towards socially close individuals in line with the *Attentional Bias Hypothesis*; [3], or due to an increased emotional transfer between socially close individuals [3]. To account for a baseline level of attention, we only included instances where the focal was close and visually oriented to a grooming interaction. To further control for attentional orientation, future studies could measure the total time the focal is oriented towards the bout. However, even this is an approximate measure that cannot convey to what degree the individual is processing the visual scene in front of them, and as chimpanzees use their peripheral vision [e.g. 68], exact orientation may not be a reliable indicator. To unpick the degree that the visual scene is processed and induces arousal, additional methodologies such as pupillometry or thermography could be used. Pairing these indicators of emotional arousal with data on varying attention levels, and studying the time profile of emotional arousal, would clarify if emotional transfer was enhanced only when attention is sustained towards any individual, or only with socially close individuals. Another explanation for this finding of a social closeness bias is that when individuals observed a close social partner grooming, this necessarily involved the presence of a preferred grooming partner, and their presence would increase the likelihood of the observer initiating grooming. Furthermore, if a chimpanzee observes their close social partner grooming another chimpanzee, this may induce

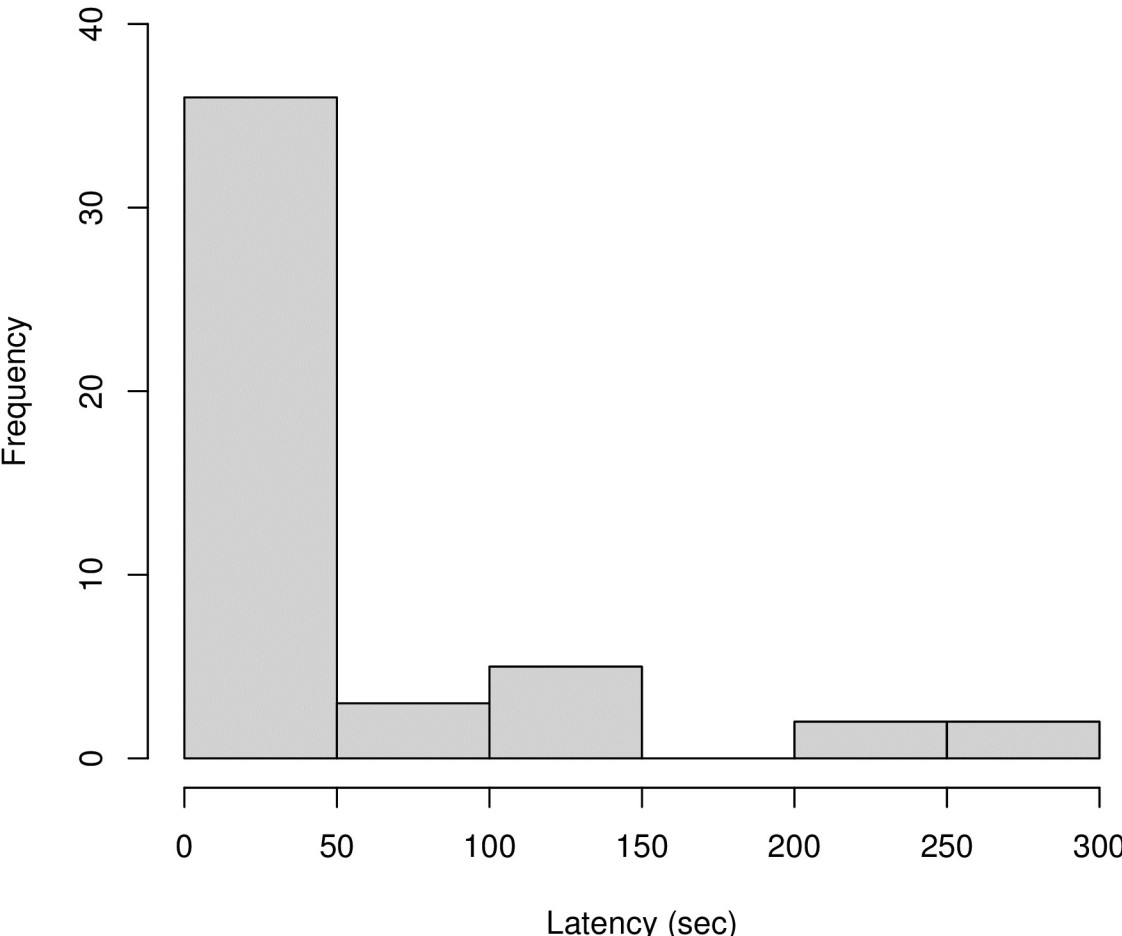

**Fig 6. Histogram of latencies from observing play to first initiating play.** Data comprised of $N = 48$ observations across $N = 28$ individuals.

a reaction akin to jealousy: chimpanzees are more likely to react negatively to social affiliation between groupmates when they have a close relationship with one of the affiliating dyad [69]. A reaction resembling jealousy may also be expressed through an increased likelihood to initiate grooming themselves. To differentiate between these explanations, we would need a larger dataset, and to include the presence of a preferred grooming partner, and the identity of the individual that the focal initiated grooming with, as additional variables.

In contrast to our hypothesis, grooming contagion was not quicker when focals observed a close social partner grooming. Whilst social closeness may influence the overall presence of contagion, the latency to catch a behaviour may be determined by factors specific to the observed or matched interactions. For example, physical proximity and grooming rates are correlated [e.g., 43], and the specific positioning of potential grooming partners may affect latency, as grooming may be more likely when partners are within arm's reach or if the independent positioning and involvement of other individuals are conducive to grooming.

Contrary to predictions that grooming contagion would be more frequent and faster in younger individuals, due to reduced inhibition [39], we found no effect of age on grooming contagion. This is most likely driven by the low frequency of younger individuals initiating grooming in the post-observation focals and at baseline; it is possible that they were influenced

in other ways after observing grooming, such as increased rates of other affiliative behaviours. In line with similar findings in studies of yawn contagion [e.g. 70]. we found no evidence for a sex effect on grooming contagion. Despite indications that males are more prone to use grooming as short-term social currency [71], we found that individuals of both sexes responded at similar tendencies and latencies. It may be the case that a sex difference lies in who the focal chimpanzee initiates grooming with after observing the behaviour; as males tend to have more stable and close social alliances [72], they may be more likely to initiate grooming with a preferred social partner, rather than with the same individuals that they observe. It may also be that distinct patterns arise when looking not just at the post-observation initiation of grooming behaviour, but also whether individuals engage in others' initiations, and the total time spent engaging in this behaviour. Further research could compare differences across these measures.

Conversely, and in support of our hypothesis, we found that younger individuals were significantly more likely to catch play behaviour, and there was a trend that they caught play faster. Contagion of interactive behaviours under voluntary control may be heightened in younger individuals due to their less developed executive function and lower inhibitory control- a pattern found in humans, apes, and macaques [38,39,73]. Alternately, play contagion may be enhanced in younger chimpanzees as this is a much more common way of affiliating for juveniles [74] and so more salient to them than grooming. Additionally, play typically involves more movement and physical disturbance of the surroundings [54], and so may attract more attention than grooming bouts, leading to an increased unsuppressed contagious response. Play contagion was not affected by social closeness; perhaps due to play bouts being more salient than grooming bouts, high levels of attention are already directed to the interaction, and so an attention bias towards socially close individuals has no additional effect.

A possible lower threshold for play contagion than grooming contagion is implied by variation in latency, whereby play was initiated significantly faster than grooming (the median latency was 18.0 seconds for play compared to 77.1 seconds for grooming). Therefore, play may be less costly for younger individuals and lower inhibition may lead to an increased likelihood to succumb to arousal initiated by observing a play bout in others. It has been suggested that inhibition is involved in contagious processes [75], but this has not been explicitly tested. Future studies may look to pair behavioural data with parallel measures of arousal to investigate whether younger individuals who catch play quickly experience greater arousal upon exposure. Emotional contagion and arousal can be measured through combining multiple modalities of behavioural and vocal observations to indicate underlying emotional states [76] whilst thermography could be used to record subtle underlying changes to underlying affective states [77–79]. Integrating physiological and behavioural measures will facilitate greater interpretation of the motivations and mechanisms that influence contagion of social behaviours among our primate relatives.

The unique patterns we observe here may reflect distinct evolutionary pressures for individuals to catch affiliative behaviours as well as negatively associated or self-directed behaviours. In socially tense situations, orangutans were more likely to catch scratching from individuals they were not socially close with, which may offer an adaptive advantage as they catch arousal allowing them to prepare for unpredictable behaviour [67]. In contrast, the tendency of chimpanzees to catch grooming from socially close individuals may endow them with an increased ability to form and maintain close social bonds. Behavioural contagion is adaptive not just as a basal foundation for empathy, but as a mechanism enabling individuals to respond and act in a context appropriate manner.

In conclusion, here we present evidence for the contagion of affiliative social behaviours in chimpanzees, widening the behavioural contagion literature. Akin to previous studies of

mimicry, contagion, and empathy, the patterns of social contagion presented here are in line with the idea that social contagion may be modulated by both bottom-up attention processes and top-down executive control. Future research should explore these processes, including more precise measures of visual attention and parallel measures of emotional arousal. Additional studies could then address the possible further variation within different types of grooming and play. Categorising interactions by emotional valence is useful when assessing general patterns in data [8,14], but considering the exact emotional profile of individual interactions may be key to fully understanding the role of affect within behavioural contagion. These patterns could serve as a model for our evolutionary ancestry, whereby sharing a sensitivity and propensity to match the affiliative behaviours of others, in addition to aversive behaviours and emotions, may have shaped our social relations and adaptive fitness.

## Supporting information

**S1 Appendix. Ethogram.**
(DOCX)

**S2 Appendix. Full results of GLMMs.**
(DOCX)

## Acknowledgments

We are very grateful to the keepers and community of Chimfunshi Wildlife Orphanage, and the Chimfunshi Research Advisory Board, for all their support. We would like to thank Roger Mundry for providing R functions used to test model assumptions and stability.

## Author Contributions

**Conceptualization:** Georgia Sandars, Zanna Clay.

**Data curation:** Georgia Sandars, Jake S. Brooker.

**Formal analysis:** Georgia Sandars, Jake S. Brooker.

**Funding acquisition:** Zanna Clay.

**Investigation:** Georgia Sandars, Jake S. Brooker, Zanna Clay.

**Methodology:** Georgia Sandars, Jake S. Brooker.

**Project administration:** Zanna Clay.

**Resources:** Zanna Clay.

**Supervision:** Jake S. Brooker, Zanna Clay.

**Validation:** Georgia Sandars.

**Visualization:** Georgia Sandars.

**Writing – original draft:** Georgia Sandars, Jake S. Brooker.

**Writing – review & editing:** Georgia Sandars, Jake S. Brooker, Zanna Clay.

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
