## [Decision Letter · Decision Letter 0]

17 Apr 2024

PONE-D-24-08243ChimpanSEE, ChimpanDO: First evidence of grooming and play contagion in chimpanzeesPLOS ONE

Dear Dr. Sandars,

Thank you for submitting your manuscript to PLOS ONE. After careful consideration, we feel that it has merit but does not fully meet PLOS ONE’s publication criteria as it currently stands. Therefore, we invite you to submit a revised version of the manuscript that addresses the points raised during the review process. As you will see, both reviewers provided positive feedback about this manuscript, while also providing concerns about the methods and the analysis. There are a number of clarifications requested regarding the PO-MC method that will need to be addressed and elaborated upon. Most notably, Reviewer #2 raises serious concerns about whether the MC selections were made appropriately to control for time of day. Reviewer #1 also raises questions about the reciprocity of grooming that does not represent contagion, while Reviewer #2 brings up a point pertaining to theoretical differences in initiating/receiving grooming. In addition, in terms of the results, both reviewers also bring up serious concerns about the discarding of the neutral pairs and whether the current Wilcoxon tests actually test the hypotheses. In addition to addressing these important issues, which address the PLOS ONE publication criteria pertaining to the technical standards and conclusions of the research, it is recommended that you also attend to the additional comments provided by the referees.

We look forward to receiving your revised manuscript.

Kind regards,

Andrew C Gallup, Ph.D.

Academic Editor

PLOS ONE

Reviewers' comments:

Reviewer's Responses to Questions

**Comments to the Author**

1. Is the manuscript technically sound, and do the data support the conclusions?

Reviewer #1: Yes

Reviewer #2: Partly

2. Has the statistical analysis been performed appropriately and rigorously? 

Reviewer #1: Yes

Reviewer #2: I Don't Know

3. Have the authors made all data underlying the findings in their manuscript fully available?

Reviewer #1: Yes

Reviewer #2: Yes

4. Is the manuscript presented in an intelligible fashion and written in standard English?

Reviewer #1: Yes

Reviewer #2: Yes

5. Review Comments to the Author

Reviewer #1: “ChimpanSEE, ChimpanDO:First evidence of grooming and play contagion in chimpanzees” is an interesting research article investigating groom and play contagion in 41 captive chimpanzees housed in a sanctuary in Zambia. The Authors used a well-suited PO-MC method to study whether chimpanzees show i) grooming/play contagion in 5 minutes after the observed grooming/play bout, and ii) whether this was dependent on their social closeness to the individual exhibiting grooming or play, or their age, sex or rank in the dominance hierarchy. The Authors compared the matched controls (MC), i.e. observations matched in time and social conditions, with the observations following grooming or playing bouts (PO). The Authors ask a timely question, use appropriate observational methods to study this question, analyse the data with modern statistical methods and discuss the findings appropriately, placing the study in the context of research on behavioural contagion and empathy. Overall, it was a pleasure to read this nice work. However, I also have some questions relating to the study design, analyses, and the interpretations of the found effects. These points are listed below in their appearing order, separated for major points, minor points and copy-edits, and I hope they will be helpful for the Authors in revising their work.

Major points.

The Authors nicely explain how the grooming or play observations were defined (“We determined whether observation happened by considering the head orientation of the observer, and their distance to the behaviour. When the trigger was within 5-metres, and happened in the 180 in front of their face and in direct visual contact, we considered the behaviour as observed. PO focals therefore started either when a grooming or play interaction started within the subject’s observational area, or when a subject moved so that they observed the behaviour.”). Thus, I assume that the focal observers were only those individuals that did not participate in the grooming or play bout themselves. However, this has not been explicitly stated in the text, so it could be that Authors also considered event participants in the analyses. If the Authors considered both the individuals that were in close proximity and observed the event (yet not included in the event themselves) and also the participants in the grooming or play event (those that did not initiate the event, but participated in it) - this is then problematic. These events are reciprocal in nature, and the receivers of grooming or play usually initiate grooming/play afterwards with the same grooming/play partner afterwards, and this would not be considered contagion. Please elaborate and if I am (hopefully) wrong, include the information that the observers did not participate in the event itself.

It is a bit confusing why the Authors only compared “attracted” and “dispersed” pairs, when in fact the “neutral” pairs made most of the pairs? How do Authors explain and deal with this big number of “neutral” pairs (e.g. L307: “N = 29 were attracted, N = 1 was dispersed, and N = 90 were neutral”, L320: N = 33 were attracted, N = 1 was dispersed, and N=62 were neutral)? This should be discussed in the text.

The review on behavioural contagion in animals (L64-74) is quite short and leaves the reader wondering about other literature. There are many more instances of contagious behaviour, including yawning, scratching, scent-marking, etc. in animals, but these are not mentioned here.

Figure 4 is not understandable. Why was the data segmented in 8 parts, and what does this Figure tell us?

Minor Points.

L20. “contagion effect” is a bit unclear, please try to rephrase.

L24. “and that it is crucial to consider behavioural contexts when determining predictors of contagious behaviour” -> which behavioural contexts and how does this statement follow from your results? Please either elaborate or delete.

L51-52. “activity reminiscent of serious functional contexts” -> do you mean that these behaviours are potentially functionally relevant for these animals?

L58-62. The play is mentioned in a non-positive context here, which leaves the reader wondering why was play then chosen as one of two ‘positive’ behaviours in this study. I would suggest moving this part to earlier in text and ending the paragraph on a more positive note about play.

First sentence of this paragraph (L76-77) should be omitted as this was already mentioned in the previous paragraph.

L79-83 are a bit broad - consider shortening and targeting more to this study.

L102, L119. To my reading, you did not measure the strength of the contagion effect, so consider rephrasing, or explain how the strength was measured.

L119-126. Would suggest rewriting this paragraph for clarity.

Data analysis. I would suggest to delete subtitles phrased as questions.

L196-203. Description of how the metrics and the dyadic sociality index were calculated is unclear, please elaborate. Furthermore, was this data later used and how?

L220-222. “we excluded all observations from that point on, as it would not be possible to determine whether the focal’s subsequent behaviour was driven by their experience observing or engaging in the behaviour.” I do not follow this sentence, could you please elaborate further, and how this impacted the duration of total time that the individual was observed for, did you control for this discrepancy in the analyses?

L225-227. Here it says that latencies were expressed as proportions of the 5-minute focal follow, but in the analyses we see the latencies expressed in seconds. Can you please explain this, and correct if/where necessary?

L253-255. These lines should be rewritten, as the mentioned variables are not predictors, but rather dependent variables in your models.

L261-262. “as some observations were recorded during the same bout of grooming or play” -> I find a bit problematic that you used same “trigger” events for multiple focal individuals, and then excluded event ID from the model, because effectively you treated these datapoints are independent which they were not. It would be important to mention how many focal observations come from the same grooming/play event in your dataset.

L301-302, L329. Consider rephrasing these questions into short statements or provide a short subtitle instead.

L330-331. I would suggest placing the models in the main manuscript, and not supplement (i.e., at least the models showing the significant social closeness and age effects).

L333 and elsewehere. A range of how many observations were done per individual is mentioned here, but in the text it is unclear what the “range” refers to; please add this information.

When reporting findings in the models, remind the reader what were the main and control predictors in the models, and what were the dependent variables.

Figure 5. Here, the differences in play are reported for age categories, but it is not clear whether the age was treated as continuous variable or as a category in the other analyses. Please add this information.

L349-351; 372-373. As the full models did not have a significantly better fit than the null model, the results need to be treated with caution.

L405-408. Would suggest omitting the part with “jealousy” as it is anthropomorphic.

Final paragraph could be shortened a bit (L466-476).

Please indicate somewhere in the manuscript that the data are made freely available, and the address where the reader can find it (this is also a requirement from the journal).

References should be sorted so that the Latin names of species are written in italic typeface, and that all journal names are capitalized. Furthermore, L557, Ref 33: Delete “Official Journal of the American Society of Primatologists”, L622-623, Ref 61: Please double-check the details as they do not seem accurately reported.

Copy-Edits.

L21, L378. Add that these were captive chimpanzees.

L22. “social closeness bias” -> delete “bias” as it is unclear

L23. “negative” -> would suggest to put it in quotation mark, or write something like “negatively valenced”

L25, L30. “foundational” -> do you mean “fundamental”?

L53. Please add “some primates” as play face is not exhibited by all primates.

L54. “solo play” -> do you mean “solitary play”?

L55. “collaboration” -> do you mean “interaction”?

L55. “social play (hereafter: play)” -> If only from hereafter it should be written as “play”, then include “social” beforehand in lines L49-55.

L95. “this suggests that this” -> “this suggests that it”

L100. “we sought to establish whether in chimpanzees there is a contagion effect for” -> consider replacing with “we sought to establish whether chimpanzees show contagion for”

L104, L135 . “hypothesis” -> “prediction”

L170. “trigger” -> perhaps exchange to “event”

L171. “in front of their face” -> “in front of the observer’s face”

L174. “all focals” -> sounds a bit colloquial, would suggest rephrasing to “all behaviour”

L183. “around them” -> “radius”

L185. “videos” -> “focals”

L187. “there were the lowest amount” -> “there was the lowest amount”

L190. “social scan observations” -> “scan samplings”

L194. “were visually recorded” -> “were recored”

L240, 241. Consider deleting “and then also” and “if not” for clarity.

L287, 289. Delete “then”

L384, L459-461. Please add quotation marks for “catch”.

L401. Delete “only”

Supplementary Appendix 1 is labelled as Supplementary Appendix 2, and Supplementary Appendix 2 is labelled as Supplementary Appendix 3; please change accordingly.

Reviewer #2: The authors studied the presence of and social behaviors that influence grooming and play contagion in chimpanzees. The topic is a valuable addition to our understanding of contagious behaviors and variables that may be related to empathy that drive these contagions. I have some serious concerns about the methods and results. With issues in the methods and results, it is hard to judge whether the discussion is appropriate until I have confidence in the data. I believe the manuscript needs a substantial revision and re-review before I can make a recommendation on its publication.

Abstract/Introduction

I’m a bit concerned about the emphasis on this being the “first” evidence of grooming and play contagion in chimpanzees, as the title reads. Other journals recommend avoiding stating results as “firsts,” and I tend to agree. Whether this is (or anything) is a first or not is often argumentative and semantic. Do contagious play faces count as play contagion? Some might argue yes. I also do not understand how auditorily induced grooming (lines 92-3, reference #33) does not count as grooming contagion in chimpnazees. Ultimately, I do not believe results need to be “firsts” to be interesting. This study is plenty compelling whether or not it is a first, so I encourage the authors to consider reframing it without the “first” element.

Methods

Line 243: With the PO-MC method, the definitions are whether the focal “initiated” the behavior in the PO or the MC. A literal reading of “initiated” has me worried about the strictness of the data. Initiated means began, so grooming/play in the MC would only be counted by the focal if the focal was not grooming/playing at the beginning of the MC window, but then started during it. Grooming/playing by the focal that commenced just before the MC that carried into the MC would not be counted. The main purpose of the PC-MC method as developed by de Waal and Yoshihara was to control for daily patterns of activity. I.e., did affiliation go up because of the conflict, or because that was the typical time of day when the animals affiliated? For example, if an MC went 10:00-10:05 and the focal started grooming at 9:58 and carried this grooming into the MC, that would not be counted in the data. This would run counter to the whole idea of controlling for time of day. It would also run counter to controlling for time of day if the authors avoided this time period for its MC because the focal was already engaged in grooming. The MC is supposed to be the exact same time of day as the PC, on the nearest subsequent day, regardless of the behaviors the individuals may already be engaged in. If the focal is already grooming/playing, that is precisely what the authors want to know, in addition to whether the behavior starts during the MC. That is the only way to control for activity changes over the course of the day. My concern comes with two elements the authors need to respond to: The first is their methods, and whether they did the MC sessions appropriately. The second is that more detail is needed in the manuscript regarding the selection of MC sessions. I am making some inferences here, so the authors need to expand on their description of their methods.

Line 245: Is it appropriate to not include grooming/playing initiated by another individual during the POs and MCs? It takes two to tango, as the saying goes. Receiving grooming/playing may reflect as much of a mood or appetite as initiating, and its that state of initiating or receiving that might be induced by watching others. An individual who does not want to be groomed or to play will simply move away. Perhaps this should be included in the analysis. ‘Grooming/playing received’ could be separated from ‘grooming/playing initiated’, and then both could be combined into ‘grooming/playing total’.

Lines 305-11 and 318-323: I’m concerned about how this analysis is done. The Wilcoxon compares attracted to dispersed, and it discards the data from the neutral pairs entirely. It is as if the neutral data do not exist. I do not know that that is appropriate. Especially as the last sentence in each paragraph reads “This indicates that individuals were significantly more likely to initiate grooming/play bouts in PO focals than in MC focals.” But, that is not what was tested. More likely to initiate in the PO than MC would include the initiations that also happened in both conditions (i.e., the neutrals). Without designating pairs as attracted, dispersed, or neutral, it would be total initiations in the PO compared to total initiations in the MC. That analysis would match the sentences in the manuscript. It seems to me that the analyses need to change to conform to the comparison the authors want to make, or else the sentences need to change to match the comparisons run.

I think what bothers me about this analysis (even if it is based on prior work) is that the authors want to know if the subjects groom/play more after viewing grooming/playing than their baseline rate of grooming/playing. The baseline rate of both is primarily housed in the neutral pairs. Throwing them out artificially lowers the amount of behavior in the control or baseline condition. The effect may still be significant, but we really should be looking at how much more grooming/playing there is relative to the baseline rate. Throwing out the neutral pairs suggests that the baseline rate of either is near 0, since there is only 1 dispersed pair in each. That magnifies the effect in the statistics (a larger Z score). Instead of what may be a small or moderate increase in grooming/playing, the statistic implies a very large increase. Unless the authors can argue otherwise, some analysis that includes the neutral pairs as part of the baseline rate is needed.

Fig 3a: I do not understand how to interpret this plot. I do not understand what the 8 partitions refers to or why they were done, and I do not understand how to visualize a difference between contagion and no contagion. There are more black dots in and around the bulge in no contagion, which implies more, but more what?

Fig 3b: I do not see a legend labeling the white and grey parts of the bars. I also don’t understand what the numbers 1-8 mean in the bars. Are those the 8 segments of data? Why was the data segmented? What does the segmentation represent? I do not see anything in the Methods related to data segmenting.

Fig 5: I do not understand why figure 5 does not match figure 3. Shouldn’t they be the same? The analysis of the grooming and contagion data was the same, so the visualization should be as well. Also, why were the grooming data segmented and the play data were not?

Discussion

Stylistic note: Sometimes when talking about one individual ‘catching’ a behavior of another, scare quotes are used and sometimes they are not. Particularly see the discussion. The authors should be consistent throughout. Either always quotes or never quotes. I also encourage the authors to find a word that does not require scare quotes to make their ideas more precise and less distracting to read.

Line 445: Is there a way to compare the latencies to groom and play statistically? If they are not statistically different, the authors should not be interpreting the medians as different.

Line 467-9: The conclusion here seems a bit strong. The authors previously speculated about attention and executive control, but now they are stating it as supported by the evidence. Since they did not directly measure attention or executive control, this sentence is overstating the results. They need to soften the language here.

6. PLOS authors have the option to publish the peer review history of their article (what does this mean?). If published, this will include your full peer review and any attached files.

Reviewer #1: No

Reviewer #2: No

---

## [Author Response · Author response to Decision Letter 0]

26 Jul 2024

We would like to thank you for the opportunity to resubmit a revised copy of this manuscript.

We are very grateful to the two reviewers who provided such helpful and constructive feedback, that we feel has significantly improved the quality of our paper. We are very grateful for all of the time and consideration that has gone into collating this feedback, and for the support of our work. We have done our best to address all of the comments and suggestions.

---

## [Decision Letter · Decision Letter 1]

23 Sep 2024

ChimpanSEE, ChimpanDO: Grooming and Play Contagion in Chimpanzees

PONE-D-24-08243R1

Dear Dr. Sandars,

We’re pleased to inform you that your manuscript has been judged scientifically suitable for publication and will be formally accepted for publication once it meets all outstanding technical requirements.

Kind regards,

Andrew C Gallup, Ph.D.

Academic Editor

PLOS ONE

Additional Editor Comments (optional):

Reviewers' comments:

Reviewer's Responses to Questions

**Comments to the Author**

1. If the authors have adequately addressed your comments raised in a previous round of review and you feel that this manuscript is now acceptable for publication, you may indicate that here to bypass the “Comments to the Author” section, enter your conflict of interest statement in the “Confidential to Editor” section, and submit your "Accept" recommendation.

Reviewer #2: All comments have been addressed

2. Is the manuscript technically sound, and do the data support the conclusions?

Reviewer #2: Yes

3. Has the statistical analysis been performed appropriately and rigorously? 

Reviewer #2: Yes

4. Have the authors made all data underlying the findings in their manuscript fully available?

Reviewer #2: Yes

5. Is the manuscript presented in an intelligible fashion and written in standard English?

Reviewer #2: Yes

6. Review Comments to the Author

Reviewer #2: The authors have done a very good job of addressing my concerns. Some of my concerns have to remain in the realm of “outside the scope of the study” or logistical limitations that do not detract from the value of the data presented. As I feel those limitations are appropriately acknowledged (where they come up), I am comfortable with the manuscript as it is constructed. The data are a valuable contribution to our understanding of behavioral contagion. With one minor correction below I support the publication of the manuscript in its present state.

Line 78: The sentence appears to be cut off.

7. PLOS authors have the option to publish the peer review history of their article (what does this mean?). If published, this will include your full peer review and any attached files.

Reviewer #2: **Yes: **Matthew W. Campbell

---

## [Editor Report · Acceptance letter]

14 Oct 2024

PONE-D-24-08243R1 

PLOS ONE

Dear Dr. Clay, 

I'm pleased to inform you that your manuscript has been deemed suitable for publication in PLOS ONE. Congratulations! Your manuscript is now being handed over to our production team.

Kind regards, 

on behalf of

Andrew C Gallup 

Academic Editor

PLOS ONE